# Exploring the Contribution of Proximal Family Risk Factors on *SLC6A4* DNA Methylation in Children with a History of Maltreatment: A Preliminary Study

**DOI:** 10.3390/ijerph182312736

**Published:** 2021-12-02

**Authors:** Francesco Craig, Eleonora Mascheroni, Roberto Giorda, Maria Grazia Felline, Maria Grazia Bacco, Annalisa Castagna, Flaviana Tenuta, Marco Villa, Angela Costabile, Antonio Trabacca, Rosario Montirosso

**Affiliations:** 1Department of Culture, Education and Society, University of Calabria, 87036 Cosenza, Italy; francesco.craig@unical.it (F.C.); flaviana.tenuta@unical.it (F.T.); angela.costabile@unical.it (A.C.); 2Unit for Severe Disabilities in Developmental Age and Young Adults, Scientific Institute IRCCS Eugenio Medea, 72100 Brindisi, Italy; mariagrazia.felline@lanostrafamiglia.it (M.G.F.); mariagrazia.bacco@lanostrafamiglia.it (M.G.B.); 30–3 Center for the at-Risk Infant, Scientific Institute IRCCS Eugenio Medea, Bosisio Parini, 23842 Lecco, Italy; eleonora.mascheroni@lanostrafamiglia.it (E.M.); annalisa.castagna@lanostrafamiglia.it (A.C.); rosario.montirosso@lanostrafamiglia.it (R.M.); 4Molecular Biology Laboratory, Scientific Institute IRCCS Eugenio Medea, Bosisio Parini, 23842 Lecco, Italy; roberto.giorda@lanostrafamiglia.it (R.G.); marco.villa@lanostrafamiglia.it (M.V.)

**Keywords:** cumulative family risk, child maltreatment, early adversity, DNA methylation, *SLC6A4*

## Abstract

The cumulative effects of proximal family risk factors have been associated with a high number of adverse outcomes in childhood maltreatment, and DNA methylation of the serotonin transporter gene (*SLC6A4*) has been associated with child maltreatment. However, the relationships between proximal family risk factors and *SLC6A4* methylation remains unexplored. We examined the association among cumulative family risk factors, maltreatment experiences and DNA methylation in the *SLC6A4* gene in a sample of 33 child victims of maltreatment. We computed a cumulative family risk (CFR) index that included proximal family risk factors, such as drug or alcohol abuse, psychopathology, parents’ experiences of maltreatment/abuse in childhood, criminal history, and domestic violence. The majority of children (90.9%) experienced more than one type of maltreatment. Hierarchical regression models suggested that the higher the CFR index score and the number of maltreatment experiences, and the older the children, the higher the *SLC6A4* DNA methylation levels. Although preliminary, our findings suggest that, along with childhood maltreatment experiences per se, cumulative proximal family risk factors are seemingly critically associated with DNA methylation at the *SLC6A4* gene.

## 1. Introduction

Childhood maltreatment includes any form of physical and/or emotional or sexual abuse, deprivation, and neglect of children, or commercial or other exploitation resulting in harm to the child’s health, survival, development, or dignity [1]. It is a global phenomenon that affects the lives of millions of children. Worldwide, it is estimated that up to 1 billion children aged 2–17 years encounter physical, sexual, or emotional violence or neglect [2]. In Italy, a national survey on child maltreatment reported that 77,000 children are victims of maltreatment, namely 9 children per 1000 residents [3]. Children who experience maltreatment often show emotional, behavioral, physical, and mental health problems [4,5,6,7] with serious life-long consequences on multiple developmental domains and functions [8].

Childhood maltreatment generally occurs within high-risk contexts characterized by several proximal family risk factors (i.e., direct and immediate caregivers’ vulnerabilities, such as drug or alcohol abuse, psychopathology, experiences of maltreatment/abuse in childhood, criminal history, domestic violence), some of which may influence both type of maltreatment and child’s outcome [9,10]. Some evidence supports the hypothesis that cumulative risk (i.e., the total number of risk factors) is a stronger predictor of adverse and detrimental child outcomes than any single risk factor [11]. Multiple risk factors in the family environment may operate conjointly to predict negative developmental outcomes [12,13,14]. Importantly, these family risk factors are not independent of one another and often cluster within the same family environment. In fact, most of the families reported for childhood maltreatment exhibit a combination of different proximal risk factors, including parent’s psychopathology, parents’ experiences of abuse in childhood, domestic violence, unemployment, poverty, and substance abuse [15,16]. Therefore, when investigating children’s bio-behavioral outcomes as a consequence of adverse events, it is important to also consider proximal mechanisms linking multiple family risk factors and the child’s health [17].

The serotoninergic system plays a key role in regulating hypothalamic-pituitary-adrenal axis (HPA) stress reactivity and its negative feedback [18,19]. Serotonin (5-HT) receptors are spread broadly throughout the central nervous system and develop early during gestation, with the serotonergic system reaching maturity during the first year of life [20]. In humans, serotonin receptors are located in peripheral tissues as well as in the central nervous system, particularly in the cerebral cortex, hippocampus, amygdala, hypothalamus, and pituitary adrenal gland [21,22,23]. These brain regions are involved in the processing of socio-emotional stress regulation and are densely innervated by serotonin neurons [24]. Altered serotonergic neurotransmission affects a wide range of neurodevelopmental outcomes (e.g., behavior, cognition, emotion, and stress regulation) [25] and psychiatric disorders (e.g., mood disorders, anxiety disorders, post-traumatic stress disorder) [26]. Feedback processes regulate the system through the serotonin transporter (5-HTT), which is encoded by the *SLC6A4* gene [27]. A rapidly growing body of research has highlighted a potential association between *SLC6A4* DNA methylation and childhood maltreatment [28]. Several studies have reported changes in *SLC6A4* DNA methylation in individuals exposed to early life adversity [25,29]. For example, higher *SLC6A4* methylation levels were found in adults exposed to parental loss and parental psychopathology [30,31]. Furthermore, childhood maltreatment, such as physical abuse [32,33], sexual abuse [29,34] or a combination of both [34], has been associated with *SLC6A4* methylation. Furthermore, increased *SLC6A4* DNA methylation has been shown in children exposed to prenatal adversity, especially those from families with an alcoholic father, indicating an early vulnerability to behavior disorders based on their reactions to stressors. [35]. Overall, although this evidence suggests that *SLC6A4* DNA methylation might be a potential biomarker even where multiple family risk factors are present. To the best of our knowledge, the relationships between cumulative proximal family risk factors, child maltreatment experiences (i.e., number of maltreatment reports), and *SLC6A4* DNA methylation remains unexplored.

In the current study, we wanted to expand upon previous findings by applying a broader conceptualization of childhood maltreatment that includes proximal family risk factors. As a consequence, in line with previous studies documenting that a cumulative index is a better predictor than any independently assessed single risk factor [36,37], we computed a cumulative family risk (CFR) index by summing a number of proximal family risk factors, such as drug or alcohol abuse, psychopathology, experiences of maltreatment/abuse in childhood, parents’ criminal history, and domestic violence [38]. In these kinds of studies, a major methodological difficulty is to establish whether infants’ *SLC6A4* methylation is affected by the severity of proximal family risk factors or by the consequences of the maltreatment experience itself. It is unclear whether the joint presence of several family risk factors and the maltreatment experience itself act as direct predictor of altered *SLC6A4* DNA methylation, or whether the relationship between CFR and altered *SLC6A4* DNA methylation is mediated by the number of maltreatment experiences experienced by the child. In light of these considerations, the aim of this preliminary study was to examine both the direct (CFR → altered *SLC6A4* DNA methylation patterns) and indirect (CFR → number of maltreatment experiences → altered *SLC6A4* DNA methylation patterns) effects of the CFR index in explaining *SLC6A4* DNA methylation patterns in child victims of maltreatment. First, we expected that a higher CFR index score would be associated with a higher number of maltreatment reports. Second, we hypothesized that both the CFR index and the number of maltreatment experiences had an effect on DNA methylation: a greater cumulative risk exposure to proximal family risk factors and a greater cumulative maltreatment experience would negatively influence (either directly or indirectly) children’s *SLC6A4* DNA methylation.

## 2. Materials and Methods

### 2.1. Participants

Participants were children with a history of maltreatment recruited at the Maltreatment Clinical Unit of Scientific Institute IRCCS “Eugenio Medea” of Brindisi (Italy). All of them had been removed from their caregivers’ care and placed in residential care by the Juvenile Justice System. Residential care refers to long-term care given to children who stay in an institutional setting rather than in their own home or family home. Children in residential care could have contact with their birth family. Such contacts were arranged by the multidisciplinary team (i.e., child neuropsychiatrist, psychology, pedagogist, social workers, and child abuse expert consultant) of the Maltreatment Clinical Unit. Children who received social services assessment because of concerns about the risk of maltreatment were not included in the study sample. Further sample exclusion criteria were the presence of any genetic abnormalities, neurosensory disabilities (i.e., blindness, deafness) or medical conditions (i.e., brain damage, autoimmune diseases, inflammatory bowel diseases, failure-to-thrive, and systemic cardiac complications).

The study design was approved by the Ethical Committee of IRCCS Giovanni Paolo II (Bari, Italy) (Protocol number 238/18). The study was also authorized by the Juvenile Justice Section (Italian Ministry of Justice) of Lecce, Italy. All of the parents or legal guardians of participants gave their written informed consent and had the right to withdraw consent at any stage.

### 2.2. SLC6A4 Methylation Assessment

One saliva sample for epigenetic analysis was taken by trained researchers at the Maltreatment Clinical Unit of the Scientific Institute IRCCS “Eugenio Medea” of Brindisi (Italy) during a neuropediatric visit. A minimum of 2 mL of saliva was obtained either by placing a salivary swab in the child’s mouth (for younger children) or by asking the child to spit into a tube (for older children). This procedure was non-invasive and stress-free. All samples were obtained after 30 min of fasting. The saliva was processed using the ORAcollect OC-175 kits (DNA Genotek, Ottawa, ON, Canada) and stored at +4 °C. Saliva samples were then delivered to the Biology Laboratory of Scientific Institute IRCCS “Eugenio Medea” (Bosisio Parini, Lecco). Genomic DNA was extracted following manufacturer’s protocols and its quality was evaluated using a Qubit fluorometer (Invitrogen, Thermo Fisher Scientific, Waltham, MA, USA). *SLC6A4* DNA methylation status was analyzed by polymerase chain reaction amplification of bisulfite-treated DNA, followed by next-generation sequencing on a NEXTSeq-500 (Illumina Inc, San Diego, CA, USA). A specific region of *SLC6A4* was analyzed: chr17:28562750–28562958, including 13 CpG sites that have been analyzed in association with childhood adversity [32,35,39]

### 2.3. Child and Family Characteristics

Information about the study participants and their families was obtained from case records using an ad-hoc data collection checklist for detecting the child’s gender and age, mother and father’s age, level of education, employment status, proximal family risk factors and child’s maltreatment history.

### 2.4. Cumulative Family Risk (CFR) Index

Proximal family risk factors, which included both maternal and paternal risk factors for maltreatment, were considered towards the CFR index. The selection of risk variables was based on pre-existing literature [16,38,40,41,42] and included: drug or alcohol abuse, psychopathology, experiences of maltreatment/abuse in childhood, parents’ criminal history, and perpetrators of domestic violence. Although unemployment and job insecurity are an additional key family risk factors [41,43], we did not include these variables in the CFR index because all parents were unemployed or had only occasional jobs, thus reducing variability. Each of the 10 variables (five for caregiver 1 and five for caregiver 2) was first categorized into a dichotomous variable: 0 = absence of risk, 1 = presence of risk. For each child, the dichotomous variables were then summed into a CFR index that potentially ranged from zero (absence of any risk factor) to 10 (presence of all risk factors in both caregivers).

### 2.5. Child Maltreatment Lived Experiences

The Maltreatment Classification System (MCS) [44] was used to assess individual children’s maltreatment experiences and code the type of maltreatment children experienced based on children’s records from child protective services. The MCS is a method for classifying maltreatment reports including dimensions of subtype, severity, frequency, chronicity, and perpetration of maltreatment. The MCS has been shown to be a reliable and valid tool to classify maltreatment in several studies [45,46]. Subtypes of maltreatment are physical abuse, sexual abuse, physical neglect (failure to provide food, shelter, clothing, hygiene, or medical care), supervisory neglect (general lack of supervision, inappropriate substitute care, or lack of supervision in a dangerous environment), emotional/psychological maltreatment, moral-legal/educational maltreatment, and drug/alcohol use. In this study, types of maltreatment were coded for all children on a dichotomous scale (0 = not reported, 1 = reported). If the record did not mention the incidence of a specific maltreatment type, this was coded as not reported. All children’s records were coded by two clinical psychologists who were blind to the aims of the paper.

## 3. Results

### 3.1. Sample Characteristics

The participants were 33 children (20 females) aged between 8 months and 15 years (M = 8.26, SD = 3.63) who had been victims of child maltreatment. Their caregivers’ sociodemographic characteristics are reported in Table 1. The mean percentage methylation for the 13 *SLC6A4* CpG sites analyzed is reported in Figure 1.

### 3.2. Maltreated Children’s Experiences and Family Cumulative Risk

Thirty children (90.9%) reported more than one type of childhood maltreatment. Figure 2 shows the percentage of children’s cumulative maltreatment experiences. These maltreatment experiences occurred over a highly variable time frame, ranging from less than one year to seven years, prior to the children being placed in residential care (M = 2.4, SD = 1.8). The prevalence of each maltreatment experience is reported in Table 2.

Prevalence of each proximal family risk factor is reported in Table 3. The mean CFR index was 2.91 (SD = 2.26), ranging from 0 to 7.

### 3.3. Preliminary Analysis

A set of correlational analyses was performed to investigate whether: (1) a greater cumulative risk exposure to proximal family risk factors (CFR) is associated with a greater number of children’s maltreatment experiences (NMRs); (2) children’s DNA methylation level at each CpG site in the promoter region of *SLC6A4* is significantly associated with both CFR index and NMRs.

The Spearman correlation revealed a non-significant association between CFR and NMRs (*r* = −0.11; *p* = 0.544). Some significant positive association between CFR index and methylation in two different *SLC6A4* CpGs sites emerged: a higher score in the CFR index was associated to higher DNA methylation at CpG5 (*r* = 0.47; *p* = 0.006) and CpG13 (*r* = 0.42; *p* = 0.016) (See Appendix A). No significant correlations were found between NMRs and *SLC6A4* DNA methylation.

### 3.4. SLC6A4 Methylation and Family Cumulative Risk

As Baron and Kenny [47], Judd and Kenny [48], and James and Brett [49] have suggested, one of the crucial conditions for mediation is that the predictor (CFR) is correlated with the mediator (NMRs). Preliminary analysis revealed no significant correlation emerged between the CFR index and NMRs. Moreover, no significant correlations were found between NMRs and *SLC6A4* DNA methylation. Thus, we tested CFR as possible direct predictors of altered *SLC6A4* DNA methylation patterns while NMRs was considered in the model as a control variable. Two hierarchical multiple regressions were performed in order to test the role of the CFR index, maltreatment and children’s characteristics in explaining the DNA methylation level at CpG5 (Model 1) and CpG13 (Model 2). For both models, the CFR index was entered into Step 1; NMRs and years since the maltreatment experience were entered into Step 2; children’s gender and age were entered into Step 3. In order to adjust for multiple testing, we used the Benjamini–Hochberg false discovery rate procedure, setting an adjusted *p*-value of 0.05 [50].

Model 1 (CpG5) was significant overall (*F* = 10.44, *p* < 0.001) and explained 67% of the variance. Similarly, Model 2 (CpG13) was significant overall (*F* = 6.87, *p* < 0.001) and explained 57% of the variance. In both models, *SLC6A4* DNA methylation was significantly explained by the CFR index, NMRs and children’s age. The higher the CFR index score and NMRs and the older the children, the higher methylation levels at CpG5 and CpG13. This association was significant. The summary and coefficients for Model 1 and Model 2 are reported in Table 4.

## 4. Discussion

The current work aimed to investigate the relationships between proximal family risk factors, child maltreatment experiences, and epigenetic patterns in child victims of maltreatment. We firstly hypothesized that a greater cumulative risk exposure to proximal family risk factors was associated with a greater number of maltreatment experiences. Contrary to our expectations, no significant correlation was found between these two variables in our sample. Although this appears to be a counterintuitive result, it should be noted that research has not provided conclusive results about this association yet. A literature review [51] investigating the association between different risk factors, including family risk factors, and the number of child maltreatment experiences showed equivocal or even contradictory findings for some specific risk factors. For example, inconsistent results were found when considering parental history of mental health problems, parental substance abuse, and domestic violence (factors also considered in the present study). Some studies did not find an association between these factors and the number of child maltreatment experiences [52,53,54]. In their review, White and colleagues [51] suggested including several factors in the risk assessment of maltreatment recurrence, such as household size, presence of grandparents or other source of family or social support, presence, and type of local services. Unfortunately, as this information was not available, we were not able to consider these variables in the CFR index. Therefore, we cannot rule out that the lack of association observed in the present study could be due to the fact that additional variables related to family functioning would have provided a more comprehensive picture of proximal family risk factors.

Our results also showed that, in children with a history of maltreatment, the severity of family risk factors, and multiple maltreatment experiences directly contributed to—at least partially—explaining the level of DNA methylation of *SLC6A4* at the CpG5 and GpG13 sites, suggesting that *SLC6A4* DNA methylation patterns might be affected by a combination of both direct (i.e., childhood maltreatment) and proximal (i.e., proximal family risk factors) variables. The strongest predictor of *SLC6A4* DNA methylation was the CFR index, which alone explained 34% of the variance in the model with CpG5 as outcome and 29% of variance in the model with CpG13 as outcome. One may wonder how these findings could explain the association between CFR and *SLC6A4* DNA methylation in maltreated children. Several child maltreatment studies suggest that proximal family risk factors are associated with high levels of parental stress [16,55] and variations in early parental care—often induced by high stress levels—result in altered neural, hormonal, cognitive, and behavioral responses in their offspring [56]. Moreover, high-risk contexts characterized by several proximal family risk factors (i.e., domestic violence, inappropriate parenting behavior) are associated with intergenerational transmission, namely parents affect their offspring’s traits through genetics, social learning or other processes [57,58], including epigenetic mechanisms [59,60]. Importantly, the CFR index range observed in this study varied between 0 to 7 (M = 2.91) suggesting that even moderate levels of family risk factors might critically contribute to altered DNA methylation patterns [30]. Thus, a possible interpretation of our findings is that DNA methylation alterations in children with maltreatment histories might be due to exposure to proximal family risk factors (not necessarily at high levels) via intergenerational transmission. Although we cannot rule out that proximal family risk factors might have affected the offspring—without necessarily invoking epigenetic explanations for intergenerational transmission—evidence from the present study suggests that, along with childhood maltreatment, proximal family risk factors would expose the child to parental stress resulting in epigenetic modifications associated with the serotoninergic system, which, in turn, plays a key role in regulating the HPA axis.

Furthermore, our results highlight the fact that a higher number of maltreatments explained a short proportion of variance in both models. A higher number of maltreatment experiences partially explain a higher DNA methylation This is in line with previous studies documenting that the cumulative experience of early childhood adversity is linked with *SLC6A4* DNA methylation [25,28,31]. Notably, one study reported that *SLC6A4* DNA methylation was significantly associated with the cumulative number of traumatic events [61]. It is important to note that the association between *SLC6A4* DNA methylation and multiple childhood maltreatment has not been consistently replicated in the literature and case-control studies offer contradictory findings of *SLC6A4* methylation between traumatized and non-traumatized impacted groups [30,62]. Nonetheless, this result encourages further exploration of the role of *SLC6A4* DNA methylation in accounting for gene-environment interactions and how it might modulate the development of psychopathology later in life.

Finally, we found that children’s age was a significant predictor of *SLC6A4* DNA methylation. Importantly, children’s age was not significantly correlated with the cumulative number of maltreatment experiences. There is much evidence that CpG dinucleotides located in the promoters of some genes become methylated with age [63,64,65]. In the pediatric population, a recent work observed an increased rate of age-associated DNA methylation changes, suggesting adjusting for age as an explanatory variable in pediatric studies of DNA methylation [63]. Consequently, the significant effect in the present study may be due to the large age range of our sample (from 8 months to 15 years). Our finding corroborates how important it is to include age as a control variable in the analysis in childhood epigenetic studies.

The present study shows some limitations. First, we focused only on a specific gene, *SLC6A4*. Nonetheless, the methylation changes observed in this gene might be detected even in other genes involved in stress regulation (for example, *NR3C1*). Further research is needed based on both a target gene approach (by considering other genes associated with early adverse events and stress regulation) and a genome-wide approach. Second, this study involved only a sample of child victims of maltreatment. It would be important to include a control sample of children without history of maltreatment to further investigate the extent of altered *SLC6A4* DNA methylation. Third, in human studies, DNA methylation markers can only be evaluated in peripheral tissues (i.e., blood, salivary sample). Nonetheless, recent findings suggest that peripheral methylation levels correlate with those measured centrally [66], with saliva samples being more closely aligned with DNA methylation patterns in the brain [67]. Fourth, considering the cross-sectional nature of our study, we cannot exclude that *SLC6A4* methylation status might have changed during the residential care stay before the assessment of DNA methylation. Therefore, future studies are warranted to employ a research design that includes different time points of DNA methylation evaluation. In addition, future studies should account for possible differences in terms of epigenetic patterns of diverse early adversities.

Notwithstanding these limitations, the current study provides preliminary evidence that epigenetic mechanisms can be potential biomarkers, not only of childhood maltreatment experiences per se, but also of proximal family risk factors. However, since at this stage it is not possible to assume causal evidence between family risk factors and DNA methylation in maltreated children, further larger-scale research focused on the association between an adverse and dysfunctional family environment and epigenetic changes in child victims of maltreatment is encouraged. A first step would be to consider the importance of clearly defining proximal family risk factors associated with specific kinds of maltreatment in well-designed epigenetic studies. Second, more research is needed to establish the role of an adverse family environment on epigenetics, including potential protective factors, and the role of epigenetic moderation informed by individual differences in child resilience. Third, it would be important to improve and expand our knowledge of the longitudinal and cumulative effects of the environment on the epigenome in children exposed to maltreatment. Finally, from a parental intervention point of view, it would be crucial to explore the impact that intervention programs for parents could have on children’s DNA methylation [68].

## 5. Conclusions

While our findings cannot prove a causal relationship between family risk factors and DNA methylation in maltreated children, the present study adds new evidence, in line with recent views, of how the family environment can become biologically embedded through epigenetic changes and can ”get under children’s skins“ [69,70]. These epigenetics vestiges remain even after the children are removed from their caregivers’ care and placed in residential care, suggesting that proximal family risk factors do play a critical role in contributing to an altered DNA methylation pattern in children. This might result in a relatively stable contribution to long-term trajectories of less-than-optimal social-emotional development in these children.

## Figures and Tables

**Figure 1 ijerph-18-12736-f001:**
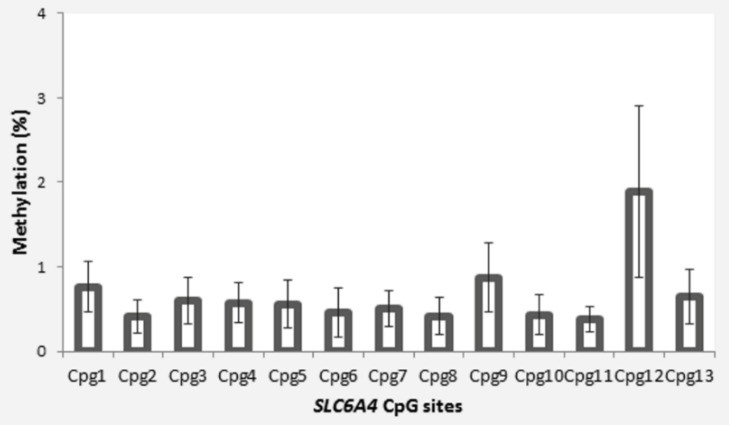
Site-specific *SLC6A4* methylation in maltreated children from our sample. (Error bars represent standard errors.).

**Figure 2 ijerph-18-12736-f002:**
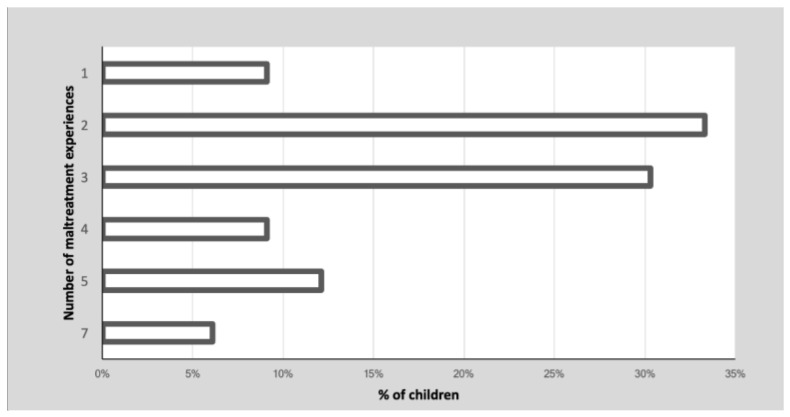
Prevalence (%) of children’s multiple experiences of maltreatment.

**Table 1 ijerph-18-12736-t001:** Parents’ sociodemographic characteristics.

	Mean, SD, Range	
Maternal age (years) (data not available from 3 mothers)	M = 38.13, SD = 7.14, Range: 27–48	
Paternal age (years) (data not available from 2 fathers)	M = 42.96, SD = 8.09, Range: 26–56	
	** *n* **	**%**
Mothers’ Level of education		
Secondary school	*n* = 24	72.7%
Not known/Not available	*n* = 9	27.3%
Fathers’ Level of education		
Professional qualification	*n* = 3	9.1%
Primary school	*n* = 2	6.1%
Secondary school	*n* = 15	45.5%
High school	*n* = 1	3.0%
Not known/Not available	*n* = 8	36.3%
Mother employment		
Unemployed	*n* = 26	78.8%
Occasional job	*n* = 7	21.2%
Father employment		
Unemployed	*n* = 25	75.8%
Occasional job	*n* = 7	21.2%
Not known/Not available	*n* = 1	3.0%

**Table 2 ijerph-18-12736-t002:** Prevalence of children’s maltreatment experiences classified according to the Modified Maltreatment Classification System.

Children’s Maltreatment Experiences	*n*	%
Emotional maltreatment	33	100
Physical neglect	29	87.9
Supervision neglect	16	48.5
Moral neglect	7	21.2
Educational neglect	5	15.2
Physical abuse	6	18.2
Sexual abuse	5	15.2

Note. The percentage sum is greater than 100 since a child may have experienced multiple types of maltreatment.

**Table 3 ijerph-18-12736-t003:** Prevalence of each parental risk variable.

Parental Risk Variable	*n*	%
Drug or alcohol abuse	23	69.7
Experiences of maltreatment/abuse in childhood	18	54.5
Psychopathology	14	42.4
Perpetrators of domestic violence	11	33.3
Criminal history	8	24.2

Note. The percentage sum is greater than 100 since there may be more than one proximal family risk factors in a single family.

**Table 4 ijerph-18-12736-t004:** Summary coefficients of regression models.

		Model 1 CpG5				Model 2 CpG13			
Steps	Predictors	*R* ^2^	*R*^2^ Change	Durbin-Watson	*β*	*R* ^2^	*R*^2^ Change	Durbin-Watson	*β*
Step 1		0.34 ***	0.34 ***			0.29 **	0.29 **		
	CFR				0.58 ***				0.54 **
Step 2		0.43 ***	0.08			0.40 **	0.11		
	CFR				0.52 **				0.47 **
	NMRs				0.29 ^#^				0.33 *
	Years since maltreatment				0.21				0.04
Step 3		0.67 ***	0.24 **	1.95		0.57 ***	0.16 *	1.62	
	CFR				0.57 ***				0.51 **
	NMRs				0.30 *				0.31 *
	Years since maltreatment				−0.13				−0.09
	Child gender				0.07				0.091
	Child age				0.50 **				0.40 *

Note. CFR = Cumulative Family Risk; NMRs = number of children’s maltreatment experiences; * *p* < 0.05; ** *p* < 0.01; *** *p* < 0.001, ^#^
*p* = 0.058.

## Data Availability

The data that support the findings of this study are available on request from the corresponding author. Moreover, the database with the 75% of the Raw Data will become available on a ZENODO repository (https://zenodo.org/).

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
