# Peer review of "Exploring the Contribution of Proximal Family Risk Factors on SLC6A4 DNA Methylation in Children with a History of Maltreatment: A Preliminary Study"

_ijerph, 2021, doi:10.3390/ijerph182312736_

Round 1

Reviewer 1 Report

The manuscript of Craig and Co-authors is well-written and very interesting.

Nevertheless, I would like to obtain some additional information, i.e.:

- Please precisely describe the characteristics of the studied group. The main limitation of the study is a broad age range of the studied children. There is probably a positive correlation between the age of children and the number of their maltreatment experiences. Could you describe the characteristics of all 33 studied children in one table? Please include the important variables: age, gender, CFR, NMRs, time when the maltreatment occurred, and the result of SLC6A4 DNA methylation.

- Have you found any differences in the SLC6A4 DNA methylation results depending on the used procedures (i.e. by taking a salivary swab from the younger children's mouth or splitting into a tube by older children)? Do you think it is possible that this factor could influence your results?

Author Response

Reviewer #1:

1-Comments and Suggestions for Authors

Comment: The manuscript of Craig and Co-authors is well-written and very interesting.

Author's response: We thank the reviewer for the positive comment.

Nevertheless, I would like to obtain some additional information, i.e.:

Comment: Please precisely describe the characteristics of the studied group. The main limitation of the study is a broad age range of the studied children. There is probably a positive correlation between the age of children and the number of their maltreatment experiences.

Author's response: The Reviewer raises an important point. Actually, we checked the correlation between the age of children and the number of their maltreatment experiences, but we have omitted this analysis in the previous version of the manuscript. The correlation between age of children and the number of their maltreatment experiences is not significant (Pearson’s correlation: r = -.037, p = 838; Spearman’s correlation: Rho = -.055, p = .770). Now we have added this information and briefly commented it in the Discussion section of the revised manuscript.

Comment: Could you describe the characteristics of all 33 studied children in one table? Please include the important variables: age, gender, CFR, NMRs, time when the maltreatment occurred, and the result of SLC6A4 DNA methylation.

Author's response: Unfortunately, due to the nature of the particular sensitive data processed and the particular type of sample, we cannot publish a table with age, gender, CFR, NMRs, time when the maltreatment occurred, and the result of SLC6A4 DNA methylation separately for each child. Indeed, at guaranteeing the individual's right to privacy, the Ethical Committee has requested to publish data in aggregated form only. In fact, insofar as improbable, it is not possible to rule out that to provide the data raw could permit to identify each child. On the other hand, we would like to point out that the present study has been partially funded by the Italian Ministry of Health, which requires the publication of the 75% of the Raw Data related to published papers in open access repositories. Therefore, if the article would be accepted for publication the database with 75% of the Raw Data will become available on the ZENODO repository and we will provide the link to it.

Comment: Have you found any differences in the SLC6A4 DNA methylation results depending on the used procedures (i.e. by taking a salivary swab from the younger children's mouth or splitting into a tube by older children)? Do you think it is possible that this factor could influence your results?

Author's response: To the best of our knowledge no study has compared potential DNA methylation difference between the two procedures in infants or children. However, it should be noted that in younger and older children saliva samples were collected using kits provided by the same company (ORAcollect OC-175 kits - DNA Genotek, Ottawa, Canada), which offer comparable performance. They allow obtaining a minimum of 2 ml of saliva either by placing a salivary swab in the child's mouth (for younger children) or by asking the child to spit into a tube (for older children). The DNA was extracted from the same biological material, i.e., saliva. In an adult study, this method provides significant amounts of high-quality and high molecular weight DNA (Langie et al., Salivary DNA Methylation Profiling: Aspects to Consider for Biomarker Identification. Basic Clin Pharmacol Toxicol. 2017). Hence, it is a reasonable expectation that the two procedures did not influence our results.

Reviewer 2 Report

The manuscript is tackling an important issue of child maltreatment and is well written. 

The study has also limitations that are discussed by the authors in the discussion section. For me, the most important limitation is number of tested subjects in combination with heterogeneity of tested sample. Therefore, the study constitutes rather a preliminary experiment warranting further investigation. I think that the authors should add an information (in title or abstract) that it is in fact a preliminary study.

The authors should also use a Spearman correlation because Pearson’s correlation can be affected by outliers leading to false conclusions. The correlations should be also presented in figures to enable readers to gain insight into the structure of the data.

Finally, I encourage the authors to provide raw data as a supplementary file or as a dataset deposited in one of many repositories, for example Mendeley Data. This is important because single studies commonly have a problem with small sample size. It is an accumulation of data that really allows for drawing firm conclusions. Therefore, sharing data will increase the impact of the study in future.

Author Response

Reviewer #2:

2- Comments and Suggestions for Authors

Comment: The manuscript is tackling an important issue of child maltreatment and is well written.

Author's response: We thank the Reviewer for the positive comment.

Comment: The study has also limitations that are discussed by the authors in the discussion section. For me, the most important limitation is the number of tested subjects in combination with the heterogeneity of the tested sample. Therefore, the study constitutes rather a preliminary experiment warranting further investigation. I think that the authors should add information (in title or abstract) that it is in fact a preliminary study.

Author's response: Following the Reviewer’s suggestion, we have described the preliminary nature of the findings in the title, abstract and manuscript. For example: “Although preliminary, our findings suggest that…” (abstract); “In light of these considerations, the aim of this preliminary study was…” (Introduction); “Notwithstanding these limitations, the current study provides preliminary evidence…” (Discussion). We also have edited the title in following way: Exploring the Contribution of Proximal Family Risk Factors on SLC6A4 DNA Methylation in Children with A History of Maltreatment: A preliminary study

Comment: The authors should also use a Spearman correlation because Pearson’s correlation can be affected by outliers leading to false conclusions. The correlations should be also presented in figures to enable readers to gain insight into the structure of the data.

Author's response: Thank you for this valuable suggestion. For the preliminary correlation, we have now used Spearman correlation instead of Pearson’s correlation. Preliminary correlations were used to identify which CpGs might be associated with the cumulative risk exposure to proximal family risk factors (CFR). Spearman correlation confirmed that SLC6A4 DNA methylation at CpG5 and CpG13 was significantly correlated with the CFR. In contrast, because Spearman correlations did not highlight significant results with respect to the association between SLC6A4 DNA methylation and number of maltreatment experiences, we used this latter variable (i.e., number of maltreatment experiences) as a control variable in the regression model. We modified the manuscript (results and discussion) considering these new analyses plan. In addition, as supplementary material, we added scatter plots related to the two significant correlations that emerged (i.e., CFR and SLC6A4 DNA methylation at CpG5; CFR and SLC6A4 DNA methylation at CpG13).

Comment: Finally, I encourage the authors to provide raw data as a supplementary file or as a dataset deposited in one of many repositories, for example, Mendeley Data. This is important because single studies commonly have a problem with a small sample size. It is an accumulation of data that really allows for drawing firm conclusions. Therefore, sharing data will increase the impact of the study in the future.

Author's response: The Reviewer raises an important point. Please note that the present study has been partially funded by the Italian Ministry of Health, which requires the publication of the 75% of the Raw Data related to published papers in open access repositories. Therefore, if the article would be accepted for publication the database with 75% of the Raw Data will become available on the ZENODO repository and we will provide the link to it.

Reviewer 3 Report

Yes (see attached file)

Author Response

Reviewer #3:

Comments:

Comment: In the introduction, at line 70 authors state that “A rapidly growing body of research has highlighted the potential role of SLC6A4 DNA methylation in childhood maltreatment”. They then report some examples from the literature. Although these examples are clear, they seem to suggest an association between SLC6A4 DNA methylation and psychopathology, maltreatment, or abuse, and not a causality between the gene’s methylation and the outcomes (as they state in lines 80-82 if I understood correctly). Rather than “role” of the gene, it would maybe be safer to change the phrasing to “link” or “association”, or in an alternative suggest the idea that epigenetic regulation may be involved in setting risk outcomes.

Author's response: We thank the Reviewer for pointing out this issue. We have edited the problematic sentence in the revised version of the manuscript according to the Reviewer's suggestion.

Comment: The age range, 8 months to 15 years, is extremely wide. The authors state in the results that SLC6A4 DNA methylation is also explained by children’s age (higher age is associated with higher methylation). In lines 308-315, the authors discuss this, but it would be interesting to delve into this a bit more. Would adjusting for age as a covariate change, in the authors’ opinion, the outcomes of the study? Which differences are expected between developmental milestones? Maybe also add a line in the limits of the study.

Author's response: The reviewer makes the following main points: (1) Would adjusting for age as a covariate change, in the authors’ opinion, the outcomes of the study? (2) Which differences are expected between developmental milestones?  

Regarding (1):  If the Reviewer is referring to covariate as intervening variable, then we should apply an analysis of covariance (ANCOVA). In this case it is possible statistically controlling for the effects of continuous variables. However, ANCOVA is not the appropriate statistical analyses for the aim of our study. For example, we did not need to compare difference between groups. On the other hand, there is a different way to qualify a variable as a potential covariate. As stated by Salkind (2010; In: Encyclopedia of Research Design, ENCYCLOPEDIA, DOI: https://dx.doi.org/10.4135/9781412961288.n85) “Similar to an independent variable, a covariate is complementary to the dependent, or response, variable. A variable is a covariate if it is related to the dependent variable. According to this definition, any variable that is measurable and considered to have a statistical relationship with the dependent variable would qualify as a potential covariate. A covariate is thus a possible predictive or explanatory variable of the dependent variable. This may be the reason that in regression analyses, independent variables (i.e., the regressors) are sometimes called covariates. Used in this context, covariates are of primary interest. In most other circumstances, however, covariates are of no primary interest compared with the independent variables.”  In our study, we have considered age as a possible explanatory of the dependent variable (as suggested by Alisch et al., 2012). In the light of above mentioned Salkind’s note, can be also considered as a covariate. Thus, it seems to us that we have already analyzed the age as a covariate change on the outcome of the study. Moreover, according to the Reviewer’s suggestion, it should be noted that we recommend the importance to consider age as a control variable in further studies. Indeed, in the Discussion section, we wrote: “Our finding corroborates how the importance important it is to include to include age as a control variable in the analysis in childhood epigenetic studies”. 

Regarding (2): This is an intriguing question. Unfortunately, we did not examine any developmental milestones. Hence it would be difficult to identify what developmental milestone should be taken into consideration. For example, previous studies have documented SLC6A4 DNA methylation is often related to emotion regulation, which could have been considered an important developmental milestone. However, a critical point is that the emotion regulation changes with age, and in our sample, the age range is extremely wide. Therefore, it could be too much speculative to establish potential differences in terms of emotion regulation during the different ages. Thus, we would prefer to avoid any speculation about this topic which could be confusing for the readers. Of course, we are available to add some consideration to this point, if the Editors consider it relevant for improving the quality of the paper. 

Minor/language and style:

  • Comment: Gene names (SLC6A4) should be in Italic, as statistic indexes (r, p)

Author's response: Fixed.

  • Comment: Line 67-68. “Feedback processes through the serotonin transporter (5-HTT), which is encoded by the SLC6A4 gene [27], regulate the system”. Not clear. Move regulate the system after feedback 2 processes? “Feedback processes regulate the system through the serotonin transporter (5-HTT), which is encoded by the SLC6A4 gene.”

Author's response: We have edited the sentence in the revised version according to the Reviewer's suggestion.

  • Comment: Line 77. “Greater early adversity family environment.” Not clear. In family’s environment?

Author's response: Thank you for noticing the mistake. Now, we have made have changed the sentence to better illuminate the topic.

  • Comment: Line 146. “Their families”

Author's response: We have corrected the above grammar mistake

  • Comment: Line 150. Blank space to be removed

Author's response: We have corrected the mistake

  • Comment: Line 160. Change “5” to “five” to be consistent with the following one

Author's response: We have edited the sentence in the revised version

  • Comment: Table 1. I guess there is a typo reporting father unemployment.

Author's response: We are sorry for the typo. We have now fixed it Table 1.

  • Comment: Line 334. “Per se” without the accent (also at line 26 in the abstract)

Author's response: Fixed.

  • Comment: Check for long sentences (e.g., 286-291).

Author's response: The new version of manuscript has been reviewed by a professional English proofreaders.

Round 2

Reviewer 1 Report

Dear Authors,

Thank you for attempting to address my concerns.